# Response of Antibiotic Resistance Genes and Related Microorganisms to Arsenic during Vermicomposting of Cow Dung

**DOI:** 10.3390/ijerph192114475

**Published:** 2022-11-04

**Authors:** Zijun Li, Chen Chen, Keqiang Zhang, Zulin Zhang, Ran Zhao, Bingjun Han, Fengxia Yang, Yongzhen Ding

**Affiliations:** 1China-UK Agro-Environmental Pollution Prevention and Control Joint Research Centre, Agro-Environmental Protection Institute, Ministry of Agriculture and Rural Affairs, Tianjin 300191, China; 2The James Hutton Institute, Aberdeen AB15 8QH, UK

**Keywords:** antibiotic resistance genes, vermicomposting, cow manure, arsenic, microorganisms, earthworms, *Eisenia fétida*

## Abstract

Antibiotic resistance pollution in livestock manure is a persistent issue that has drawn public attention. Vermicomposting is an ecofriendly biological process that can render livestock manure harmless and resourceful. However, little is known about the impact of vermicomposting on antibiotic resistance in livestock manure under stress caused by potentially toxic arsenic levels. Herein, lab-scale vermicomposting was performed to comprehensively evaluate the shift in antibiotic resistance genes (ARGs) and related microorganisms in fresh earthworm casts as well as vermicompost product health (i.e., nutrient availability and enzyme activity) when they were fed on arsenic-contaminated cow manure. The results showed that the earthworms’ interaction with cow dung led to a significant reduction in ARG concentrations, especially for tetracycline ARGs (*tet*-ARGs), β-lactam ARGs (*bla*-ARGs), and quinolone ARGs (*qnr*-ARGs). However, arsenic significantly enhanced ARG accumulation in earthworm casts in a dose-dependent manner. Moreover, vermicomposting increased the percentage of Bacteroidota in the converted products. Furthermore, arsenic exposure at low concentrations promoted the proliferation of Proteobacteria, whereas high concentrations had little effect on Proteobacteria. Our study provides valuable insight into the changes in the antibiotic resistome and related microorganisms during vermicomposting of arsenic-amended cow manure, and it is crucial to explain the environmental impact of earthworms and improve our understanding of the reciprocal benefits of soil invertebrates.

## 1. Introduction

In animal nutrition, arsenic, similar to selenium, molybdenum, iodine, and fluorine, although an essential trace element in the animal body, is also a potentially toxic one. Arsenic has been widely used in livestock and poultry breeding for a long time because of its ability to promote animal growth and improve feed efficiency [1]. However, excess arsenic is excreted through animal manure, and since it is not biodegradable, prolonged use would certainly cause irreparable damage to the environment and even its accumulation in the food chain, posing serious threats to human health [2]. Notably, the presence of excessive arsenic, caused by deep-seated pollution effects, has developed into molecular-ecological pollution, causing disruptions of proteins, DNA, enzymes, and other biological macromolecules at the molecular level, further causing carcinogenesis, teratogenesis, and mutagenesis in animals and humans. Arsenic adversely affects both the survival and metabolism of soil animals [3]. Studies have shown that anthropogenic arsenic contamination alters the bacterial community of the earthworm gut and affects functional symbiotic communities and intestinal nutrient cycling [4]. In addition, combined pollution of arsenic and Polymyxin B can enrich ARG abundance in soil and earthworm gut microbiotas [5].

Widespread use of veterinary antibiotics fuels the contamination of livestock waste with ARGs, which seriously threatens animal safety and human health [6,7,8,9]. Different from traditional chemical pollutants in the environment, ARGs can be transmitted vertically through bacterial proliferation and horizontally through the exchange of genetic information between bacteria and can spread around the world in this way [10,11]. Notably, heavy metals and antibiotics can induce cross-resistance, and thus, bacteria can become resistant to both. Toxic metals are assumed to exert selective pressure on metal-resistant bacteria, and genes associated with heavy metals, such as arsenic and cadmium, are frequently found in plasmids associated with ARGs [12,13]. Another study found that arsenic stress increases ARG abundance and dissemination risk [5]. Complex contamination by antibiotics and heavy metals poses a significant threat to ecosystems and human health through changes in the diversity and composition of soil microbiome [14] or through co-selection pressure on microbial antibiotic resistance [15].

Vermicomposting is a biochemical decomposition process of livestock waste that involves the interaction between earthworms and microorganisms [16]. Vermicomposting is a more harmless and resourceful reduction process than regular composting. Earthworms can alter the composition and structure of microorganisms in the environment [17,18] and have a significant effect on ARGs by affecting bacterial distribution [19]. Remarkably, earthworms, such as annelids, use surface respiration and are relatively sensitive to pollutant stimuli [20]. Selective stress on the local microbiome in the earthworm gut through the ingestion of food containing mixed contaminants, such as antibiotics and various toxic metals, maintains a favorable ecological niche for the colonization of antibiotic-resistant bacteria (ARB). The composition and numbers of ARB communities in the earthworm gut may increase under the long-term effects of this selection. Thus, long-term selection of antibiotic resistance in the earthworm gut may lead to ARG spreading from the earthworm gut to soil, plants grown in the soil, and ultimately to humans exposed to ARGs [21]. Therefore, it is necessary to fully reveal the influence and contribution of heavy metals on ARGs in the process of vermicomposting of cow dung to prevent and control ARG pollution and its spread in the process of animal conversion.

In this study, *Eisenia fetida* was chosen as the experimental organism to study the effect of arsenic on ARGs and their host bacteria during vermicomposting via lab-scale vermicomposting with the addition of arsenic to livestock manure (i.e., cow dung). The objectives of this study were to: (1) determine the effect of earthworms (*Eisenia fetida*) on ARGs in cow manure by investigating the growth of ARGs in cow manure with the presence of earthworms, (2) explore the response of ARGs and related microorganisms to arsenic cow dung during vermicomposting at two concentration levels, and (3) identify the main factors influencing the variation in ARGs in the process of vermicomposting.

## 2. Materials and Methods

### 2.1. Materials

Mature specimens of *Eisenia fetida* with a well-developed clitellum were used as the subject earthworm species. Trisodium arsenate (Na_3_AsO_4_·12H_2_O) was purchased from Shanghai Chemical Reagent Purchasing and Supply Station. The earthworms were purchased from Tianjin Baiming Technology Development Co., Ltd. (Tianjin, China) Cow dung was collected from a dairy farm in Tianjin, China. Before the experiment, to improve the palatability of earthworms to cow manure, the cow dung was naturally stacked for 30 days to decompose.

### 2.2. Experimental Design

Nine parallel vermicomposting reactors were set up that were randomly divided into three groups, each with three replicates. According to the actual residual range of arsenic in cow dung, the supplemental concentrations of arsenic in the three groups were set at 0, 10, and 50 mg/kg, and the groups were named CK, T_10_, and T_50_, respectively. Based on 1 kg cow manure and different arsenic concentration treatments (0, 10, or 50 mg/kg), the corresponding Na_3_AsO_4_ was weighed and dissolved in 600 mL deionized water. Then, the solution was sprayed evenly on the cow dung, which was stirred while spraying. Fermented cow dung (1.0 kg) was then added to each reactor. The reactor was an earthworm breeding box with dimensions of 28 cm × 19.5 cm × 11.5 cm and was drilled on the bottom to remove excess lixiviate. To prevent the earthworms from escaping, the top of the box was covered with gauze. About 24 h before exposure, the worms were removed from the cultures, rinsed with water, and kept on damp filter paper in darkness at 20 °C to obtain fresh casts. After depuration, 300 clitellum-banded *Eisenia fetida* of similar weight were placed in each plastic box. The experiment was conducted at room temperature (22–24 °C), and distilled water was sprinkled once every 3 days to keep the moisture content at 65–75%. The experimental period lasted 14 days. Before and after vermicomposting, the earthworms and the conversion product were collected from each reactor for measurement. Earthworms were counted every two days, and their weight was measured at the beginning and after vermicomposting of the experiment.

### 2.3. Chemical Properties Analysis

The obtained conversion product was air-dried naturally. The dry samples were ground and sifted through a particle size of 20 mesh (0.85 mm) for subsequent chemical analyses. All chemical analyses were based on Chinese organic fertilizer standards (NY/T 525-2021). First, 5.0 g of air-dried sample was added to 50 mL of deionized water, and the pH of the solution was measured at 25 °C with a pH meter. Next, organic matter (OM) was measured according to the instructions contained in the Soil Organic Matter (SOM) Kit (Shanghai Zeye Biotechnology Company Limited, Shanghai, China) based on the potassium dichromate volumetric method [22]; the total nitrogen (TN) content was determined by the Kjeldahl method, 1883 [23]. Furthermore, total phosphorus (TP) content was determined by the vanadium molybdenum yellow colorimetric method by spectrophotometry (HACH/DR5000, USA). The activity of superoxide dismutase (SOD), catalase (CAT), and peroxidase (POD) were measured according to the instructions of the enzyme activity kit (Beijing Solarbio Science & Technology Co., Ltd., Beijing, China).

### 2.4. DNA Extraction from Various Samples

The earthworms were removed and placed in a dark, damp petri dish for 24 h, and the intestinal contents were emptied and retained. DNA samples from earthworm intestinal contents were extracted using the FastDNA Spin Kit as per manufacturer’s instructions (MP Biomedicals, Santa Ana, CA, USA). The same method was used to extract DNA from the cow manure and earthworm casts. The obtained DNA samples were resuspended in DNase/RNase-free water, and then the DNA concentration was determined by Qubit 2.0 fluorescence meter (Life Technologies, Carlsbad, CA, USA). All DNA extracts were stored at −20 °C before analysis.

### 2.5. ARGs and 16S rRNA Quantification

The expression of 21 specific subtypes of ARGs (*sul*1, *sul*2, *tet*L, *tet*X, *tet*Q, *tet*O, *tet*W, *bla*_ampC_, *bla*_TEM-1_, *bla*_OXA-1_, *bla*_NDM_, *aad*A, *fex*A, *cfr*, *erm*C, *erm*B, *qnr*B, *qnr*S, *oqx*B, *str*A, *str*B), two key integrase genes (i.e., *intl*1 and *intl*2), and the 16S rDNA gene (characterizing the total amount of bacteria) in samples was quantified by real-time qPCR. Additional details of the primers used are provided in the Appendix A. Real-time qPCR analysis was performed on a 7500 Real-Time PCR System (Applied Biosystems). qPCR reactions were performed in 96-well plates, and the reaction mixture (20 μL) contained 0.4 μL of ROX Reference DyeII, 0.4 μL of each primer, 6.8 μL of DNA-free water, 10 μL of SYBR Premix Ex Taq TM II (TliRNase H Plus, Takara), and 2 μL of DNA template. The real-time qPCR conditions were as follows: an initial cycle at 95 °C for 30 s, followed by 95 cycles at 95 °C for 5 s and 40 cycles at 60 °C for 34 s, as previously described [24].

### 2.6. 16S rRNA Gene High-Throughput Sequencing

The V3-V4 hypervariable region of the bacterial 16S rDNA gene was PCR-amplified using primers 338F (5′-ACTCCTACGGGAGGCAGCAG-3′) and 806R (5′-GGACTACHVGGGTWTCTAAT-3′) in a thermocycler system. PCR amplification was performed using the following program: denaturation at 94 °C for 5 min, followed by 30 cycles of 30 s at 94 °C, annealing at 50 °C for 30 s, elongation at 72 °C for 60 s, and a final extension at 72 °C for 7 min. PCR amplification was performed in triplicate, with 25 μL reaction mixture containing 12.5 μL of 2× Taq Plus Master Mix, 3 μL of BSA (2 ng/μL), 1 μL of each primer (5 μM), 7.5 μL of ddH2O, and 30 ng of template DNA. The resulting PCR products were extracted from a 2% agarose gel and further purified. Purified amplicons were pooled in equimolar amounts and paired-end sequenced (PE 300) on an Illumina MiSeq platform (Illumina, San Diego, CA, USA) at Allwegene Technology, Beijing, China.

### 2.7. Statistical Analysis

The abundance and mean values of the target genes were calculated using Microsoft Excel 2020. Heat maps and histograms were plotted using OriginPro 2020 software (OriginLab, Northampton, MA, USA). Alpha diversity indices, including observed_species, chao1, and Shannon, were calculated using QIIME (Version 1.7.0) and displayed with R software (Version 4.1.2), as well as statistical analysis by one-way Tukey. The qvalue software package of R software (Version 4.1.2) was used to analyze the species difference at the phylum level. RDA and other plots were constructed using R software (version 4.1.2). Student’s *t*-test and ANOVA were conducted by SPSS software (IBM SPSS Statistics, USA) for multiple comparisons to determine statistical significance at the *p* = 0.05 level.

## 3. Results and Discussion

### 3.1. Evaluating Changes in Earthworm Population and Enzyme Activity during Vermicomposting

In general, earthworm survival is closely related to their density, habitat environment, and food quality [25]. As shown in Figure 1, earthworm survival showed a slightly decreasing trend for all three treatments (Figure 1a), but the net weight increased in all cases (Figure 1b). Although the survival rate of the control group was slightly higher and the weight gain was slightly greater, there was no statistically significant difference between the control groups and arsenic treatment groups (F = 0.705, *p* > 0.05), indicating that the addition of arsenic did not have a significant effect on earthworm survival, which is consistent with Tang’s study [26]. The decrease in earthworm population may be due to the selection of heavier earthworms in this study, their higher initial density, and the differences in individual earthworms as they adapt to their new environment.

Microorganisms in the vermicompost products produce catalase, which breaks down hydrogen peroxide and is usually indicative of the size or activity of the vermicompost products [27]. Therefore, peroxidase activity is an indicator of the total activity of stable enzymes in the vermicompost products matrix and the activity of the current live flora in these casts. Changes in peroxidase and catalase levels in the different treatment groups before and after vermicomposting are shown in Figure 2. The passage of the residue through the intestine of earthworms increased the activities of peroxidase and catalase in earthworm feces. Compared with the control group, the increase in the POD and CAT enzymes in the earthworm model after arsenic treatment was lower than that in the control group (Figure 2). Catalase activity may be an important indicator of the degree of soil bioremediation [28]. The increase in redox enzymes indicates a turnaround in environmental conditions, and after vermicomposting, the earthworm manure environment quality improved compared with the environment without earthworms, indicating the restorative effect of earthworms on the environment, which is in line with Wu’s study [27].

Oxidative stress is thought to be one of the main processes of arsenic-mediated toxicity; thus, oxidative damage biomarkers and antioxidant enzymes have been widely used to evaluate the ecotoxicity of pollution [29]. Dramatic changes in the oxidoreductase activity were also observed in the earthworm body (Figure 3). In comparison with the control group, the low concentration of arsenic did not significantly affect peroxidase activity in earthworms (t = −1.39, *p* > 0.05), but the high concentration of arsenic resulted in a greater decrease in peroxidase activity in earthworms (t = 6.174, *p* < 0.05). In contrast, the activity of peroxidase increased after conversion compared with that before conversion, but the increase was much lower in the arsenic treatment group than in the control group. Although the difference between high and low concentrations was not significant, the catalase activity was slightly lower under high stress than under low stress. After cow dung conversion, SOD activity was higher in earthworms under low arsenic stress and lower in earthworms under high arsenic stress compared with that in the control group. Excessive accumulation of reactive oxygen species in organisms leads to lipid peroxidation (LPO) and DNA damage [30]. Antioxidant defense systems can easily neutralize any form of toxic effect or repair damage caused by reactive oxygen species. SOD is one of the main antioxidants involved in the production of superoxide anions through dismutation [31]. Malondialdehyde (MDA) is one of the final metabolites of the peroxidation reaction, in which free amino acids of proteins can react in response to cellular damage. The activity of elevations in SOD and catalase enzymes suggests that exposure initially causes cellular damage and then activates the antioxidant defense system to reduce this damage and subsequently reduce the antioxidant response, which is consistent with Deng’s observations [32].

### 3.2. Vermicomposting Reduced Some ARG Abundance and Mobile Genetic Element (MGE) Marker Genes in Cow Manure

In the present study, a total of 21 ARGs and two main MGE marker genes were analyzed in the substrate that was collected before (the initial samples) and after (vermicompost products) vermicomposting and fresh earthworm casts. The occurrence and variation in ARGs during vermicomposting are shown in Figure 4. The results showed that the absolute abundance of most ARGs in the vermicompost products was reduced to different degrees compared with that in the initial cow dung, indicating that earthworms had a reducing effect on some antibiotic resistance determinants, which is consistent with the results of Han et al. [33]. Furthermore, sulfonamide resistance genes (*sul*-ARGs) were most abundant in the initial substrate of cow dung, followed by tetracycline resistance genes (*tet*-ARGs) (Figure 4), which may be due to the widespread use of sulfonamides and tetracyclines in animal breeding [34].

After vermicomposting, a decrease in the absolute abundance of most ARGs was observed in vermicompost products. Among the ARGs, *bla*-ARGs had the highest removal rate (23.4%). Vermicomposting also had removal effects on *bla*_ampC_, *oqx*B, *bla*_NDM_, and *qnr*B, with removal rates ranging from 20.61% to 43.49% by vermicomposting. These results showed that vermicomposting could reduce ARG abundance in cow manure, similar to the results of Cui et al. [35]. Previous studies have shown that ARGs have different antibiotic resistance mechanisms; for instance, tetX can undergo enzymatic modifications [36], and *tet*M encodes ribosomal protective proteins [37,38], indicating that the products of these ARGs have different fates. Notably, all of the ARGs determined were present in the fresh earthworm casts of earthworms fed with cow dung (Figure 4), with the highest absolute concentration of *sul*1 at 3.4 × 10^7^ copies/g; second, the concentration of the class I integrase gene *intl*1 was as high as 2.6 × 10^6^ copies/g. However, it is now clear that the concentration of most ARGs in the fresh earthworm casts is generally lower than the abundance of the corresponding ARGs in earthworm casts over the same period. Furthermore, the majority of ARGs in the fresh earthworm casts are derived from feed microbes. The decrease in ARGs abundance after gastrointestinal passage of earthworms may be due to the loss and demise of ARG-containing bacteria, which further confirms the role of earthworms in reducing ARGs in the environment. Based on the study by Yang et al., it can be hypothesized that earthworm-associated symbiotic strains enhance the degradation of antibiotics and, thus, weaken the selection pressure of soil microorganisms to some extent, thus reducing the abundance of ARGs in the substrate [39].

Earthworms play a key role in organic decomposition, and epigeic earthworms directly influence the organic matter of soil decomposition through processes associated with the gut, that is, through the uptake, digestion, and absorption of organic matter and micro-organs, which are then released in earthworm feces [40]. The earthworm gut is considered a bioreactor, where the intake of cow dung undergoes major physical and biochemical changes [41,42]. The anaerobic environment in the earthworm gut could make it difficult for aerobic bacteria harboring ARGs from cow dung substrate to survive [43]. Predation by earthworms reduces the abundance of ARGs by eliminating bacteria in the living substrate, curbing their horizontal transfer [35]. Additionally, as one of the most important mobile genetic elements, integrons offer the possibility of the horizontal transfer of ARGs between bacteria [44,45]. ARG transmission depends primarily on the HGT of MGEs (e.g., plasmids, integrons, and transposons) [46]. Compared with untreated cow dung, the absolute concentrations of intl1 and intl2 decreased by 3.58% and 4.56%, respectively, after earthworm treatment. This suggests that vermicomposting can reduce the abundance of integrons and the potential risk of ARG transmission [18,47].

### 3.3. Arsenic Enriched ARGs in Vermicompost Products and Fresh Earthworm Casts

The absolute abundance of ARGs in cow manure showed significant changes upon the addition of arsenic and earthworms, as shown in Figure 5 and Figure 6. After vermicomposting, the total abundance of ARGs in the arsenic treatment group increased by 0.11- (T_10_) and 0.13-fold (T_50_) compared with that in the control group, which indicated that the addition of arsenic could induce the proliferation of ARGs in cow manure. There was a significant decrease in the absolute concentrations of *bla*_TEM-1_, *oqxB*, *bla*_ampC_, and *bla*_NDM_ in the T_10_ treatment, with decreases of 31.86%, 25.99%, 22.77%, and 21.77%, respectively. Compared with untreated cow dung, the absolute concentrations of *cfr* and *ermC* increased by 16.88% and 12.99% after vermicomposting, respectively. In contrast, in the T_50_ group, there was a significant decrease in the absolute concentrations of *oqxB* and *bla*_TEM-1_ by 24.27% and 18.01%, respectively, and the concentration of *tet*X increased by only 8.64%. This result suggests that the response of different ARGs varies with the introduction of arsenic. The total absolute abundance of the T_10_ and T_50_ treatments increased significantly after vermicomposting compared with that in the control treatment. In addition, class 1 and class 2 integrons, as one of the most important MGEs, play an important role in the transfer and transmission of ARGs. Compared with untreated cow dung, the absolute concentrations of *intl*1 and *intl*2 increased by 8.87% and 3.80%, respectively, in the T_10_-treated group and increased by 6.90% in the T_50_-treated group. Furthermore, the absolute concentration of *intl*2 decreased by 1.84% compared with the control group. This demonstrates that the addition of As could increase the abundance of integrons, which might also accelerate the potential transmission risk of ARGs [48].

Overall, the absolute concentration of most ARGs decreased through vermicomposting, and the ARGs in the wormcast were more abundant than those in the fresh casts of earthworms (Figure 6). Furthermore, as seen in Figure 6, the reduction in ARGs in the cow manure was greater in the high-concentration treatment group than in the low-concentration treatment group. The concentration of ARGs in the T_50_ group showed a greater decrease than that in the T_10_ treatment group. In the T_10_ and T_50_ treatment groups, the removal rate of nine and eleven ARGs, respectively, was higher than 10%. The removal efficiency of *bla*_TEM-1_ was the highest in the T_10_ treatment group, whereas that of *oqx*B was the highest in the T_50_ treatment group after earthworm treatment. β-lactam ARGs are known to pose a high risk to human health, and their removal can be improved at low concentrations of As contamination, which also provides a new direction for reducing the environmental risk of such ARGs. In the assay of wormcasts, it was found that the ARG concentration in the treatment group under high arsenic stress was generally lower than that in the treatment group with low arsenic addition. Studies have shown that arsenic has a toxic effect on earthworms [49], the severity of which increases with increasing arsenic concentration, thus reducing their transformation metabolism.

Overall, the presence of As contamination affected the reduction in ARGs during vermicomposting, with the absolute concentration of ARGs in the As treatment group in the substrate decreasing at a much lower rate than that in CK without As addition, and the absolute concentration of ARGs in fresh wormcasts was also generally higher than that in CK (Figure 7). However, the dominant gene concentrations remained the same, including *sul*-ARGs, *tet*-ARGs, and *str*-ARGs.

Notably, arsenic has a toxic effect on earthworms, and the reduction in ARGs by earthworms in the treatment group with added arsenic is diminished by the weakening of metabolic functions [50]. In addition, the concentration of ARGs has been shown to be significantly correlated with As concentrations [51], as it can promote the proliferation of ARGs in the environment [48]. The distribution of ARGs is largely influenced by the structure of the bacterial communities [52]. The addition of high concentrations of arsenic resulted in a greater decrease in the concentration of ARGs compared with the reduction in the treatment group with low concentrations of arsenic, which contradicts the findings of Zhao et al. [51]. It is assumed that the concentration of ARGs was reduced to some extent because of the toxicity of high concentrations of arsenic to microorganisms carrying ARGs. Heavy metals are difficult to degrade in the environment, which can exert continuous selective pressure on ARGs in the environment.

### 3.4. Earthworm and Arsenic Treatment Reshaped Microbial Compositions in Vermicompost Products and Fresh Earthworm Casts

In the present study, 5073 operational taxonomic units (OTUs) were generated from all samples, and 4752 remained after leveling. Chao and Shannon indices are often regarded as ideal indicators of bacterial community richness and diversity, respectively [20]. Figure 8 shows that regardless of the Chao and Shannon indices, the microbial richness in earthworm feces was obviously higher than that in fresh earthworm casts, and the Chao and Shannon indices indicated that the wormcast had a higher diversity of microorganisms with more OTUs. From the comparison before and after vermicomposting, the results showed that the richness and diversity of microbial strains in vermicompost products increased significantly after vermicomposting.

Figure 9 shows the variation in microbial species in different samples at the phylum level. As can be seen from the figure, there are differences in microorganism diversity in the vermicompost products and fresh casts. After vermicomposting, the microorganisms in the wormcasts consisted mainly of Proteobacteria (29.34%), Actinobacteria (29.15%), Bacteroidota (15.63%), and Chloroflexi (8.64%). The microorganisms in fresh casts were mainly Proteobacteria (51.44%), Actinobacteria (22.42%), Bacteroidota (10.19%), and Firmicutes (8.71%). After vermicomposting, the change in Bacteroidota in CK (without As addition to the substrate) was more pronounced, increasing from 8.61% to 12.76%. The proportion of Bacteroidetes increased significantly after the conversion. Bacteroides is a very diverse phylum of bacteria that is believed to exist in all parts of the mammalian gastrointestinal tract despite differences in pH, nutrients, and oxygen availability [53]. Additionally, there was a significant decrease in the percentage of *Firmicutes* after vermicomposting.

In the low arsenic treatment group, the percentage of Bacteroidota in the substrate increased significantly, increasing from 9.45% to 16.20%, suggesting that Bacteroidota is resistant to arsenic. This was followed by Proteobacteria, which decreased from 33.37% to 28.66%. The low concentrations of arsenic contamination promoted the proliferation of Proteobacteria, whereas high concentrations did not have a significant effect on Proteobacteria. This was more prominent in fresh wormcasts of all treatment groups contaminated with arsenic; there were significantly higher percentages of Proteobacteria and Bacteroidota and lower percentages of Actinobacteria and Chloroflexi. However, this trend was more evident in the low-concentration treatment group rather than in the high-concentration one. The relative abundances of Bacteroidetes and *Aspergillus* in substrates exposed to arsenic and fresh earthworm feces increased significantly, which is similar to the findings of Wang et al. [54]. Treatment with a low concentration of arsenic obviously increased the number of Proteobacteria in the earthworm gut, implying that arsenic exposure leads to the deterioration of the fresh cast microbiota, which is consistent with the findings of Li et al. [5]. However, the change in Proteobacteria under high concentrations of arsenic stress was not significant compared with CK, and the difference in microbe composition in fresh casts was greater between the groups treated and not treated with arsenic. In addition, Deferribacterota appeared in wormcasts contaminated with added arsenic after vermicomposting, presumably as a stress-responsive microorganism of arsenic in the environment. Moreover, Abditibacteriota appeared in wormcasts after earthworm transformation, which is presumed to be a microorganism specific to the earthworm transformation process.

Numerous studies have shown that the lower diversity and abundance of bacteria in wormcasts than in the surrounding soil is the result of a unique gut microenvironment [55,56]. Most microbiological activities associated with earthworms may be caused by ad hoc communities that experience selective environmental stimuli during transport. Earthworms affect the decline in soil microbial diversity but increase microbial activity through their contribution to transitional communities [57]. The effect of arsenic exposure on gut microbes was greater than that on soil microbes. Based on microbial composition analysis, arsenic exposure significantly reshaped the microbial community structure in the environment and fresh earthworm casts. The translocation of organic matter through the intestinal tract of earthworms strongly alters the structure and activity of the microbiome, revealing the direct influence of earthworms on bacterial and fungal microbial decomposers.

In addition, network analysis was performed, showing correlations between four phyla, Actinobacteria, Bacteroidota, Proteobacteria, and Acidobacteriota, because these four gut bacteria are richer and/or more closely related than others. As the dot size indicates the count of bacteria, the more bacteria there are, the larger the dot size. Similarly, a line connecting two dots represents a connection between two bacteria. The thicker the line, the greater the correlation (Figure 10). As shown in Figure 10, *Aeromonas*, *Acinetobacter*, and *Flavobacterium* were the three most abundant bacteria, but they were mainly positively correlated only with Bacteroidota and Proteobacteria, indicating the importance of these three bacteria in earthworm metabolism. In contrast, Acidobacteriota and Actinobacteria were positively correlated with other microorganisms. Thus, Acidobacteriota and Actinobacteria play a more important role in maintaining overall bacterial metabolism.

### 3.5. Driving Forces of the Changes in ARG Abundance during Vermicomposting

To further elucidate how microbial characteristics and environmental factors during vermicomposting affect the fate and behavior of ARGs, a redundancy analysis was performed. As shown in Figure 11, the first and second axes explain 42.72% and 26.10% of the selected variables, respectively. pH, DOC, and TP were positively correlated with the RDA1 axis, with the degree of influence being TP > pH > DOC. This suggests that the feed structure of the animal may influence the ARG of the cattle manure samples [58]. It can also be observed that the pH of the environment has a strong influence on the conversion efficiency of earthworms, and pH changes may inhibit the self-replication process of microorganisms, thus impacting the spread of ARGs [59]. Degradation of organic matter also had a great influence on ARGs, and the correlation between various types of ARGs and microbial abundance was different, as shown in Figure 11, which could be due to the different host bacteria encoding each ARG.

As shown in Figure 11, the pH and DOC were strongly correlated with *bla*-ARGs. Furthermore, pH was significantly positively correlated with qnr-ARGs. The microbial community abundance index (Shannon index) for *erm*-ARGs was extremely high, indicating that *erm*-ARGs were mainly carried and spread by microorganisms in the environment.

In order to identify the possible ARG hosts, the Spearman correlation between ARG-like levels and bacterial communities was investigated, as illustrated in Figure 12. A total of 15 of the 35 classes were observed to be positively correlated with most target genes, suggesting that these classes may be potential hosts for ARGs. These classes included Methylomirabilota, Acidobacteriota, Sumerlaeota, MBNT15, Planctomycetota, Chloroflexi, Gemmatimonadota, Myxococcota, Dependentiae, Entotheonellaeota, RCP2-54, SAR324_clade_Marine_group_B, Latescibacterota, Armatimonadota, and NB1-j. The genome of Acidobacteria members contains several genes associated with active molecules, which play an important role in the transmission of antibiotic resistance [60]. However, due to a lack of understanding of the physiological properties of Acidobacteria, the mechanism by which earthworm gut digestion reduces ARG is not clear. In addition, it is unclear how the nutritional metabolic pathway of Chloroflexus participates in the biochemical cycles of C, N, S, and other elements. These elements are closely related to the spread of ARGs [61].

## 4. Conclusions

(1)Vermicomposting reduced the abundance of most ARGs in cow dung and changed the structure of microbial communities in the related environment.(2)Arsenic residues in cow dung enriched ARG abundance and also reshaped microbial compositions in vermicompost products, significantly increasing the number of Bacteroides and Proteus.(3)The MGEs exhibited a trend similar to that of ARGs during vermicomposting of cow manure, which might be explained by the synchronous shift in ARGs and MRGs.(4)In the vermicomposting process, changes in physicochemical factors in cow dung played a direct or indirect role in ARG variation, and TP was the dominant factor driving the proliferation of ARGs.

## Figures and Tables

**Figure 1 ijerph-19-14475-f001:**
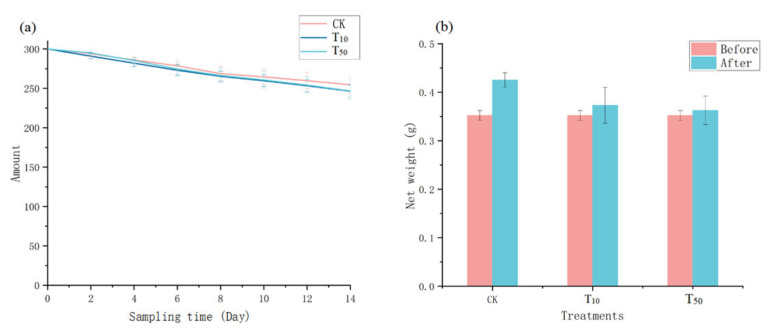
The number of living earthworms (**a**) and their weight changes (**b**). Error bars indicate standard deviations (*n* = 3). CK, T_10_, and T_50_ represent the treatment groups with 0, 10, and 50 mg/kg arsenic added, respectively.

**Figure 2 ijerph-19-14475-f002:**
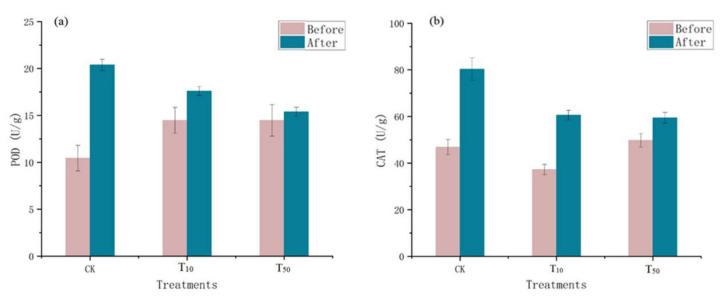
Changes in peroxidase (**a**) and catalase (**b**) in wormcasts before and after vermicompost. CK, T_10_, and T_50_ represent treatment groups with 0, 10, and 50 mg/kg arsenic added, respectively.

**Figure 3 ijerph-19-14475-f003:**
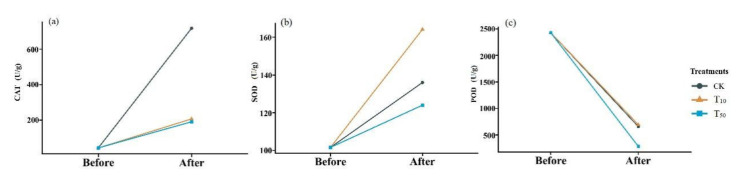
Changes in catalase (**a**), superoxide dismutase (**b**), and peroxidase (**c**) in earthworms before and after conversion. CK, T_10_, and T_50_ represent treatment groups with 0, 10, and 50 mg/kg arsenic added, respectively.

**Figure 4 ijerph-19-14475-f004:**
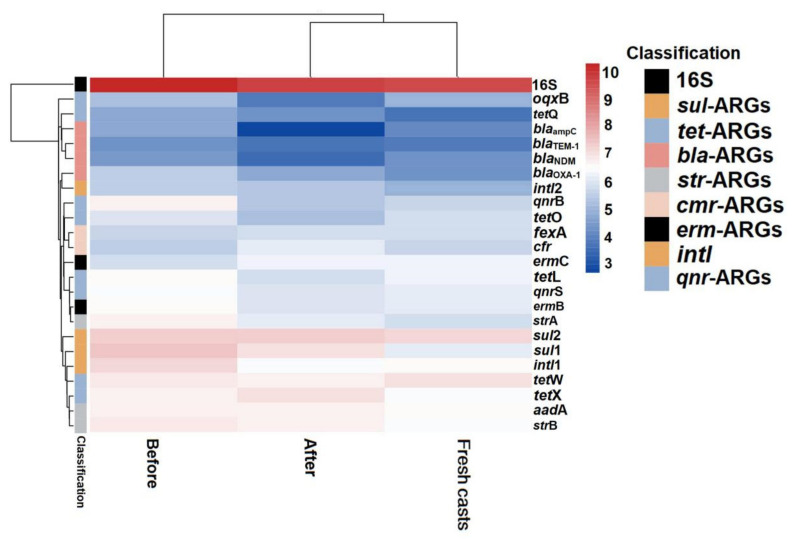
Concentrations of resistance genes before and after transformation of earthworms without added pollutants.

**Figure 5 ijerph-19-14475-f005:**
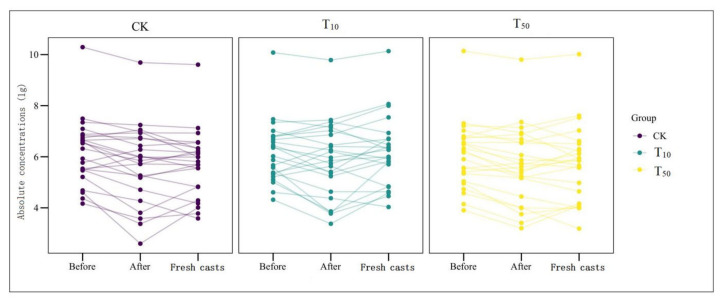
Changes in absolute concentrations of ARGs before and after vermicomposting in different treatment groups: CK, earthworm feeding on cow dung without arsenic addition; T_10_, earthworm feeding on cow dung with 10 mg/kg arsenic; T_50_, 50 mg/kg arsenic, after digestion for 14 days.

**Figure 6 ijerph-19-14475-f006:**
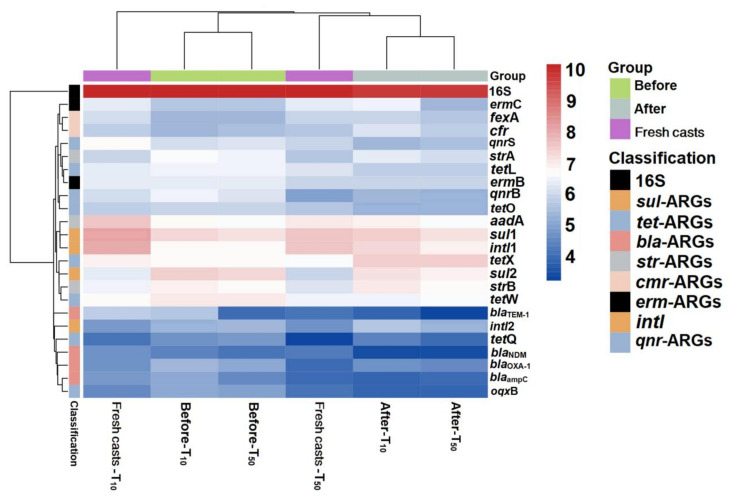
Heat map of changes in absolute abundance of ARGs in fresh casts and before and after vermicomposting under different concentrations of arsenic. CK, T_10_, and T_50_ represent treatment groups with 0, 10, and 50 mg/kg arsenic added, respectively.

**Figure 7 ijerph-19-14475-f007:**
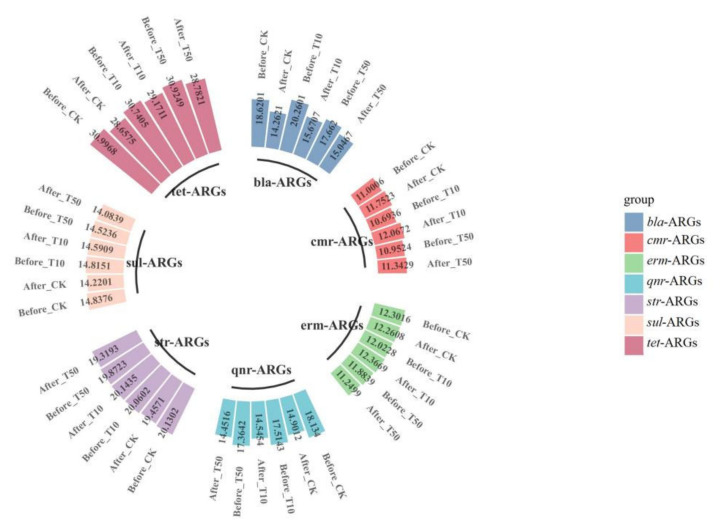
The absolute concentrations (lg) of various genes in fresh casts and before and after vermicomposting. The concentrations of resistance genes in fresh casts and products before and after vermicomposting were compared in the pollution treatment without and with different concentrations of arsenic.

**Figure 8 ijerph-19-14475-f008:**
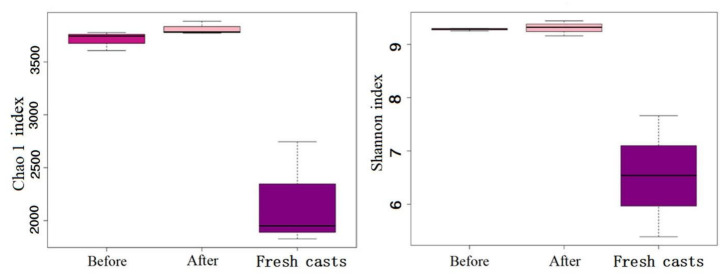
Alpha diversity of microbial communities in fresh casts and before and after vermicomposting. Chao1 index reflects the richness of bacteria, and Shannon index reflects microbial diversity.

**Figure 9 ijerph-19-14475-f009:**
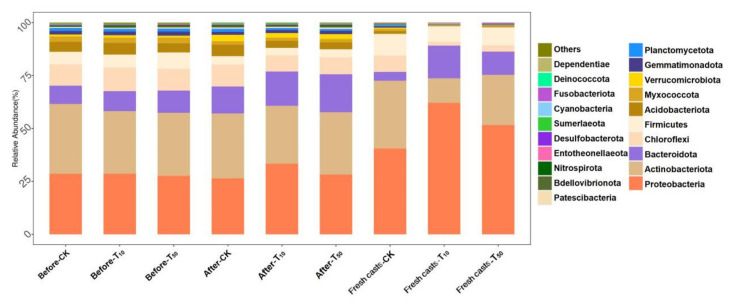
The differences between substrate and fresh earthworm feces at the phylum level. Horizontal coordinates are the names of samples, and vertical coordinates are the relative abundance of species in samples. The figure shows information on species with relative abundance of 1% or more. CK, T_10_, and T_50_ represent treatment groups with 0, 10, and 50 mg/kg arsenic added, respectively.

**Figure 10 ijerph-19-14475-f010:**
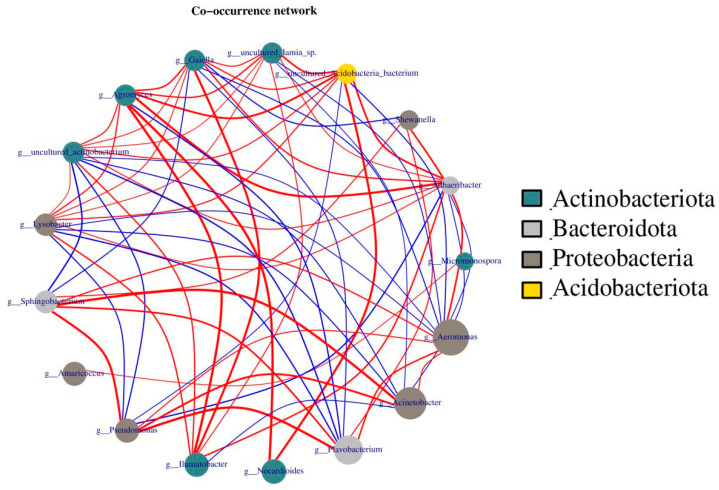
Analysis of microbial networks in cow dung and vermicompost products. The size of the dots represents abundance, the thickness of the lines represents correlation, the colors of the dots represent the phylum to which they belong, the red lines represent positive correlation, and blue lines represent negative correlation.

**Figure 11 ijerph-19-14475-f011:**
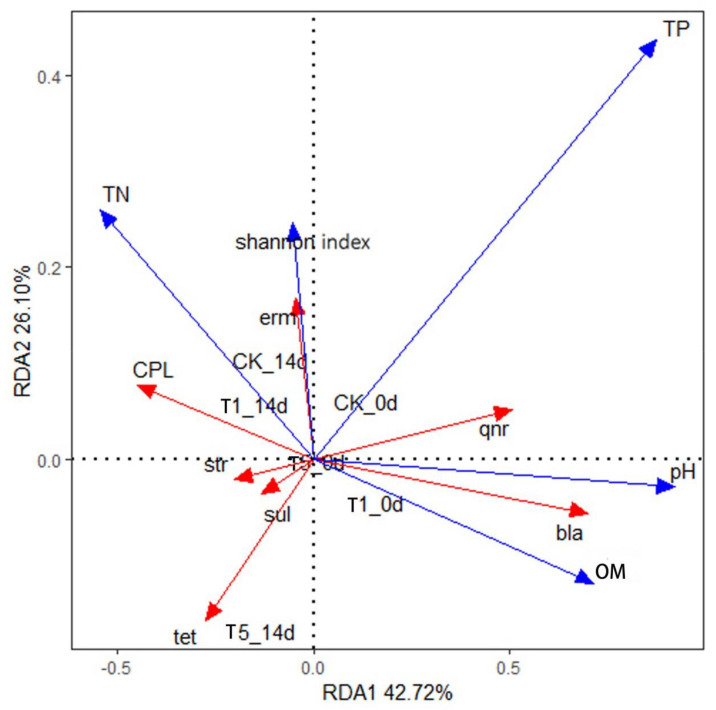
Redundancy analysis (RDA) of resistance genes, microorganisms, and environmental factors in earthworm transformation in cattle manure. a indicates day 0 of conversion, and b indicates day 14 of conversion. Abbreviation: OM, organic matter; TP, total phosphorus; TN, total nitrogen; Shannon index represents diversity of microbial communities; 0 d represents before of vermicomposting; 14 d represents after vermicomposting.

**Figure 12 ijerph-19-14475-f012:**
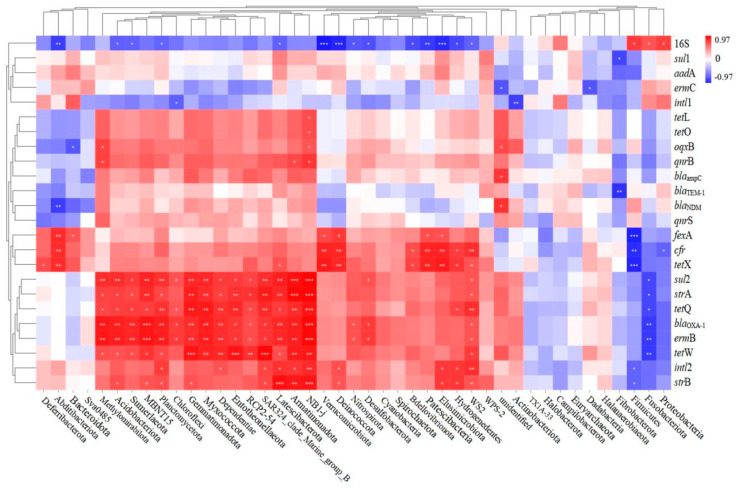
Spearman correlation analysis of target genes (ARG, *intl*1, *intl*2) with bacterial communities (Phylum levels).

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
