# Peer review of "Response of Antibiotic Resistance Genes and Related Microorganisms to Arsenic during Vermicomposting of Cow Dung"

_ijerph, 2022, doi:10.3390/ijerph192114475_

Round 1
Reviewer 1 Report
More replications should be done to extract valid conclusions. It would be necessary to increase the number of replicates to make the work reproducible.
The terminology used with regard to earthworms is not adequate.

Reviewer 2 Report
In this investigation, authors selected Eisenia fetida as the experimental organism to study the effect of arsenic on ARGs and their host bacteria during earthworm conversion via lab-scale vermicomposting with the addition of arsenic to livestock manure. This is an important investigation, not only crucial to explain the environmental impact of earthworms, but also give insights on the reciprocal benefits of soil invertebrates. Overall, this in a well written manuscript, this reviewer only has minor suggestions.
L89-95: Some descriptions on earthworm were repeated.
L114: To ensure moisture content, biomass products are usually dried in a temperature-controlled oven. Why the obtained conversion product in this investigation was air-dried naturally.
L470: The “conclusion” section is difficult to read. It is recommended to organize the major points in a list manner.
Round 2
Reviewer 1 Report
I still think that the number of replicates is too small to obtain reliable statistical results. But this can no longer be changed.
When the text indicates that the results are statistically different, the statistical result obtained should be clearly indicated, e.g. (F = xxxxx; p<0.05) if an ANOVA has been performed or (t = xxxx; p<0.05) if the statistical analysis has been a t-student.
I suggest including these indications in the text
